# Outcomes tested in non-pharmacological interventions in mild cognitive impairment and mild dementia: a scoping review

Elyse Couch 🔘 , Vanessa Lawrence, Melissa Co, Matthew Prina

Health Service and Population Research, King's College London, London, UK

**Correspondence to**
Elyse Couch;
elyse.couch@kcl.ac.uk

## ABSTRACT

**Objectives** Non-pharmacological treatments are an important aspect of dementia care. A wide range of interventions have been trialled for mild dementia and mild cognitive impairment (MCI). However, the variety of outcome measures used in these trials makes it difficult to make meaningful comparisons. The objective of this study is to map trends in which outcome measures are used in trials of non-pharmacological treatments in MCI and mild dementia.

**Design** Scoping review.

**Data sources** EMBASE, PsychINFO, Medline and the Cochrane Register of Controlled Trials were searched from inception until February 2018. An additional search was conducted in April 2019

**Eligibility** We included randomised controlled trials (RCTs) testing non-pharmacological interventions for people diagnosed with MCI or mild dementia. Studies were restricted to full RCTs; observational, feasibility and pilot studies were not included.

**Charting methods** All outcome measures used by included studies were extracted and grouped thematically. Trends in the types of outcome measures used were explored by type of intervention, country and year of publication.

**Results** 91 studies were included in this review. We extracted 358 individual outcome measures, of which 78 (22%) were used more than once. Cognitive measures were the most frequently used, with the Mini-Mental State Examination being the most popular.

**Conclusions** Our findings highlight an inconsistency in the use of outcome measures. Cognition has been prioritised over other domains, despite previous research highlighting the importance of quality of life and caregiver measures. To ensure a robust evidence base, more research is needed to highlight which outcome measures should be used over others.

**PROSPERO registration number** CRD42018102649.

## INTRODUCTION

Delivery of both pharmacological and non-pharmacological treatment in the early stages of dementia has been identified as a global priority.[1][2] Current pharmacological treatments for the cognitive symptoms of dementia have been found to have a greater effect when

### Strengths and limitations of this study

► This scoping review has systematically mapped which outcome measures have been used by randomised controlled trials testing non-pharmacological treatments in mild dementia and mild cognitive impairment.

► This review has explored how the use of outcome measures varies by diagnosis, type of intervention, country and year of publication.

► The papers included in this review were limited to full randomised controlled trials, other study designs may be using different types of outcome measures.

► Further research is needed to establish which measures should be used over others.

delivered as early as possible.[3] However, the benefits of delivering non-pharmacological treatments early are less well understood. Non-pharmacological treatments are an important clinical tool for managing dementia as they are more acceptable to some and less prone to side effects, making them a safe alternative to drug treatments.[4] Those diagnosed earlier in the disease have more cognitive abilities available to engage with non-pharmacological treatments and bolster their own methods for coping with the disease.[5] Previous systematic reviews have found non-pharmacological treatments can improve outcomes; however, these reviews were restricted to a small number of outcome measures.[6][7]

Mild cognitive impairment (MCI) has been identified as a potential prodrome for dementia, with approximately 10% of people with MCI converting to a diagnosis of dementia per annum.[8] There is an interest in MCI, as a diagnosis of MCI can facilitate an early diagnosis of dementia and therefore earlier access to dementia services and treatment.[9] MCI is a potentially reversible condition, with many people with MCI

reverting back to normal levels of cognition.[9] Therefore, it is important treatments are available. However, it is not clear which treatments can reverse MCI or prevent conversion to dementia.[3] No drug treatments for MCI have been found to be effective[10 11] and acetylcholinesterase inhibitors are not recommended, however, there is some limited evidence that non-pharmacological interventions may be beneficial.[3 12]

Randomised controlled trials (RCTs) testing non-pharmacological treatments in dementia and MCI are becoming more common. However, they are highly heterogeneous in terms of participants recruited, quality of the study and the types of interventions they are testing, making it difficult to establish the effectiveness of one treatment over another.[6 12 13] Compounding these issues is the inconsistent use of outcome measures in this area of work.[9 14]

Systematic reviews have identified possible benefits of non-pharmacological treatment, yet meta-analyses are difficult to conduct due to the variation in outcome measures used by studies and typically yield small-to-moderate effect sizes.[6 7] It is possible that these small effect sizes are due to the selection of outcome measures which either lack sensitivity or the change following the intervention not being in the area covered by the outcome measure. It is important researchers are clear on which domains their interventions are targeting, and which measures are best able to capture this change.[15] Pharmacological treatments target specific biological pathways underlying the disease; therefore, outcome measures have been chosen to reflect this and typically focus on cognitive and functional decline.[16] Non-pharmacological treatments generally do not target the underlying biological pathway of the disease therefore, outcome measures should theoretically differ between pharmacological and non-pharmacological treatments.[17] However, a review on non-pharmacological approaches to treating found that studies tended to pay little attention to the mechanisms of change underlying the intervention.[4] The expected mechanisms of change should affect which outcomes are used in non-pharmacological treatments for mild dementia and MCI.

In addition to being clear on how change arises in non-pharmacological treatments, there needs to be a more coherent use of outcomes and the measures used to capture these between studies to ensure a broad and robust evidence base.[15] In 2008, the INTERDEM group, a consortium of dementia researchers across Europe, did work to draw a consensus on which outcome measures should be used when evaluating non-pharmacological treatments. They recommended 22 measures across 9 domains including quality of life, mood, global functioning, behaviour, daily living skills, caregiver mood, caregiver burden and staff morale.[15] This guidance does not explore outcomes by the stage of the disease. The outcome measures were selected based on their applicability to European research. The utility of outcome measures may vary by culture,[16] previous reviews exploring the use of outcome measures in dementia research have not investigated how this differs by country.[17]

It is not understood which outcome measures are currently being used in non-pharmacological treatments for early dementia and MCI. Scoping reviews present the opportunity to map the evidence on a topic,[18] unlike a systematic review scoping reviews can be used to summarise the evidence in a heterogeneous body of literature. Therefore, the aim of this scoping review is to map trends in which outcome measures are being used in RCTs for non-pharmacological treatments in MCI and mild dementia.

### Objectives
The specific objectives of this scoping review are to:
1. Chart which outcomes measures have been used to assess the effectiveness of non-pharmacological treatments in mild dementia and MCI.
2. Highlight which types of measures have been used most frequently.
3. Explore whether the outcome measures used differ depending on the type of intervention, study population and country the research was conducted in.

### METHODS
#### Protocol registration
The protocol for this review was developed following the guidelines set out by the Preferred Reporting Items for Systematic Reviews and Meta-Analysis Extension (PRISMA) statement[19] and the PRISMA guidelines for Scoping Reviews.[18]

#### Eligibility criteria
We included RCTs testing non-pharmacological interventions for people diagnosed with MCI or mild dementia. Studies were restricted to full RCTs; observational, feasibility and pilot studies were not included.

Studies were included if they met the following criteria:
► Testing non-pharmacological interventions. Studies were not excluded if participants were also treated with acetylcholinesterase inhibitors.
► Participants had a diagnosis of MCI or mild dementia, which was either diagnosed in clinical practice, or met standardised diagnostic criteria, such as the International Statistical Classification of Diseases or The Diagnostic Statistical Manual of Mental Disorders, The National Institute of Communicative disorders and Stroke and the Alzheimer's Disease and Related Disorders, the International working group on MCI criteria, The Consortium to Establish a Registry for Alzheimer's Disease, The National Institute on Aging-Alzheimer's Associating Diagnostic Guidelines for Alzheimer's Disease, the Petersen Criteria; or was defined by a standardised clinical measure, such as scores between 24 and 18 on the Mini-Mental State Examination (MMSE); scores ≤26 on the Montreal Cognitive Assessment, scores between 15 and 27 on the St Louis University Mental Status, a Clinical

Dementia Rating score of 1 (for dementia) or 0.5 (for MCI); or a 4 (for dementia) or 3 (for MCI) on the Global Deterioration Scale. Studies which include a mix of participants with early dementia and MCI were included, however, studies which included healthy participants and participants with dementia at the later stages of the disease were excluded.

► The intervention was targeted for the person living with dementia or MCI. Dyadic interventions, interventions delivered to both the person living with dementia and their caregivers, were included. Interventions delivered solely to caregivers or healthcare professionals were excluded.

► Participants were living in long-term care facilities or the community.

► Written in English.

Studies were excluded if:

► Only pharmacological interventions were tested.

► The participants were diagnosed with vascular cognitive impairment, young-onset dementia, Parkinson's disease dementia or MCI with Parkinson's disease.

► Participants were living in a psychiatric inpatient or acute hospital setting.

► The intervention had the primary aim of treating major depressive disorder.

► The study tested palliative care interventions or advanced care planning.

► The only outcome measures used were economic outcomes, such as cost-effectiveness, etc.

### Information sources and search strategy

To identify potentially relevant studies, we searched EMBASE, PsychINFO, Medline and the Cochrane Register of Controlled Trials from inception until 22 February 2018. An additional search was conducted on 2 April 2019. See online supplementary table 1 for the final search strategy for MEDLINE, which was adapted for the other databases. The final search results were exported into EndNote where duplicates were removed.

Additional papers were identified by searching the references of included papers and other systematic reviews. Conference abstracts and publications were not included.

### Selection of sources of evidence

Study selection was managed in Rayyan, where citations were screened against the inclusion and exclusion criteria. Rayyan is an online app for systematic reviews which allows researchers to create their own coding system for decision making.[20] References were first screened by title and abstract, followed by a full-text screening. A second reviewer (MC) screened 10% of the articles at each stage of the review. Disagreements were resolved by discussions with a third reviewer (AMP).

A critical appraisal or assessment of the risk of bias is not necessary for a scoping review.[18] This scoping review is not aiming to critically appraise the cumulative literature of outcome measures for non-pharmacological treatment in MCI and mild dementia, therefore we did not conduct a critical appraisal or risk of bias assessment for this review.

### Data charting process and data items

Data from eligible studies were charted using a standardised extraction tool designed for this study. Items deemed most relevant to the review objectives were the diagnosis of the study participants, description of interventions being tested, the number of intervention groups and outcome measures used with references.

### Synthesis of results

The charted data were mapped to reflect the objectives of this review. Following data charting, outcome measures which were used more than once across the included studies were grouped by domain. We grouped the interventions thematically by the type of intervention being tested.

We explored which types of outcome measures were used by intervention type, by tabulating the type of intervention against the domain of the outcome measure. We excluded interventions which were only used once from this summary. Results were presented in tables and summarised narratively.

### Patient and participant involvement

The South London and Maudsley MALADY group, of current and former carers of people living with dementia, were consulted in the planning of this study.

## RESULTS
### Included studies

After duplicates were removed, a total of 7056 citations were screened for inclusion, 653 were screened at full text and 74 papers were initially identified. A top-up search in April 2019 identified 119 new citations, 17 were included making the total number of included studies 91 (figure 1).

The studies included in this review are described in table 1, including diagnosis of included participants, number of intervention groups, details on the interventions and comparisons tested and the number of outcomes measures used. The included studies were published between 2002 and 2019.

The majority of studies included in this review were conducted in the USA (n=10), Hong Kong (n=10) and Italy (n=11), followed by mainland China (n=7), Japan (n=8), South Korea (n=8) and Canada (n=6). Studies were also conducted in: Argentina, Australia, Brazil, Czech Republic, Denmark, France, Finland, Germany, Greece, Hungary, Iran, Norway, Pakistan, Singapore, Spain, Taiwan, The Netherlands, Turkey and the UK; these countries had fewer than five included studies each.

Most studies only recruited participants with MCI (n=71), followed by mild dementia only (n=14), and six

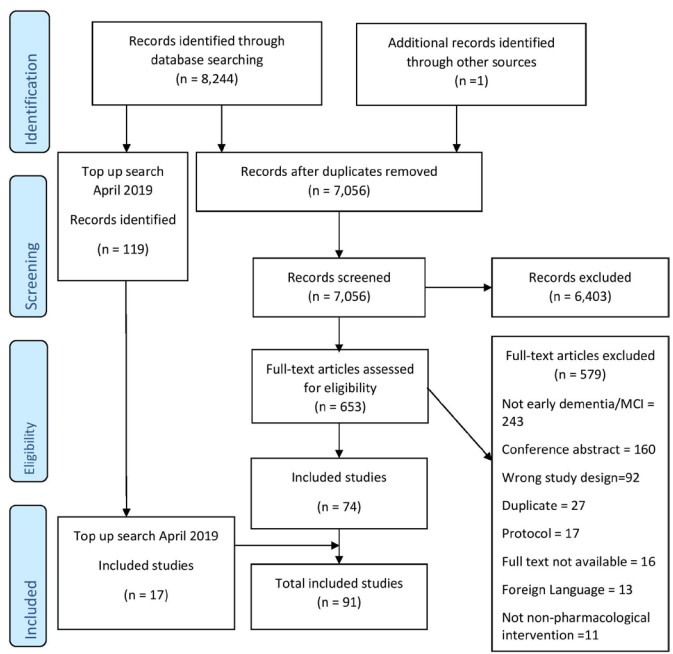

**Figure 1** Flow chart of included studies.

studies recruited both participants with MCI and mild dementia.

### Results of individual sources of evidence

We extracted 358 individual outcome measures from the included studies, of these 78 (22%) were used more than once. Out of the 78 measures used more than once, 70 (88%) were measures of participants living with dementia (PLWD), 6 measures were used in both the PLWD and their caregiver, 2 measures were only of the caregiver. The number of outcome measures used by each study ranged between 1 and 21 with an average of 6.85.

### Types of non-pharmacological interventions

We grouped the interventions thematically by type. The most frequently tested type of intervention was cognitive training (n=37) followed by physical activity (n=25), combined physical activity and cognitive training (n=4), multicomponent psychosocial interventions (n=4) and support groups (n=3). Animal-assisted therapies, art-based therapies, case management, Chinese calligraphy, music-based interventions and reminiscence therapy were each tested in two studies.

A group weight loss programme, mindfulness, social activities, transcranial direct current stimulation, transcutaneous electrical nerve stimulation and Transcranial magnetic stimulation were each trialled once. These interventions were not included in the analysis of trends in outcome measures.

### PLWD outcome measures

Table 2 presents the PLWD-specific outcome measures grouped by domain. The most frequently measured domain in PLWD was cognition/memory, which was measured 219 times across the 93 included studies. The most frequent measure of cognition was the MMSE,

which was measured 37 times. In addition to measures of memory performance, knowledge of memory strategies was measured 3 times in PLWD.

The next most frequently measured domain in PLWD was behavioural and psychological symptoms of dementia (BPSD), within this depression was the most commonly measured BPSD. The Geriatric Depression Scale was the most used measure in this domain, followed by the Neuro-psychiatric Inventory which examines a greater number of symptoms. Other BSPDs measured were apathy and agitation resulting from memory problems.

Quality of life and well-being were measured 15 times across the included studies. Quality of life was measured 15 times using four different instruments, the most popular of which was Logsdon's Quality of Life in Alzheimer's disease scale which was used 7 times.

Measures of everyday living, physical ability, biological outcomes and adherence to the intervention delivered in the study were measured <20 times across the included studies.

### Caregiver measures

Eight interventions in this study were dyadic,[21–28] all included outcome measures specific to the caregiver in addition to the PLWD. One study of an intervention solely delivered to the PLWD also included a caregiver-specific measure.[29]

Table 2 also presents the outcome measures administered to caregivers grouped by domain. The Center for Epidemiological Studies Depression Scale and the Zarit Caregiver Burden interview were the only measures which were administered solely to caregivers. The other caregiver measures were also administered to PLWD. The most frequently measured domain in caregivers was depression, followed by caregiver burden. General well-being, knowledge of memory strategies, quality of life and stress were each measured once.

### Use of outcome measures over time

RCTs of non-pharmacological treatments in mild dementia and MCI have become more frequent over recent years. Almost half (48%) of studies included in this review were published between 2016 and 2018.

Figure 2 charts trends in outcome measure domains over time. As the number of studies in this area has increased over time, so too has the use of outcome measures in all domains. Cognition/memory has consistently been measured over other domains from the beginning of this sample. The only noticeable trend change is in measures of BPSD, which was generally in line with other domains until around 2012, when it overtakes other domains.

Nearly all studies in 2014 included a measure of everyday living; however, since then, the number of studies including these measures has declined. Where measures of everyday living are being used less, measures of BPSD are being used more.

Similarly, caregiver measures were consistently used until 2011, when in 2010 and 2011 all studies included

**Table 1** Included studies

| Study | Country | Diagnosis | Number of groups | Group 1 | Group 2 | Group 3 | Group 4 | Group 5 | Number of measures |
|---|---|---|---|---|---|---|---|---|---|
| Amjad et al[37] | Pakistan | MCI | 2 | Aerobic exercise | Non-aerobic exercise | – | – | – | 4 |
| Bae et al[38] | Japan | MCI | 2 | Multi-intervention programme | Active control | – | – | – | 10 |
| Baker et al[39] | USA | MCI | 2 | Aerobic exercise | Stretching | – | – | – | 11 |
| Belleville et al[40] | Canada | MCI | 3 | Cognitive training | Psychosocial intervention | Control | – | – | 7 |
| Biasutti and Mangiacotti[41] | Italy | MCI | 2 | Cognitive training | Gym activities | – | – | – | 4 |
| Bono et al[42] | Italy | MCI | 2 | Animal assisted therapy | Control | – | – | – | 4 |
| Burgio et al[43] | Italy | MCI | 2 | Numerical training | Executive training | – | – | – | 13 |
| Buschert et al[44] | Germany | MCI | 2 | Cognitive training | Active control | – | – | – | 5 |
| Carretti et al[45] | Italy | MCI | 2 | Cognitive training | Active control | – | – | – | 16 |
| Cavallo et al[46] | Italy | Dementia | 2 | Cognitive training | Active control | – | – | – | 3 |
| Chan et al[47] | Hong Kong | MCI | 2 | Chinese calligraphy | Computer activities | – | – | – | 13 |
| Chan et al[48] | Hong Kong | MCI | 2 | Chinese calligraphy | Computer activities | – | – | – | 8 |
| Choi and Lee[49] | South Korea | MCI | 2 | Ground kayaking | Home exercise education | – | – | – | 7 |
| Combourieu Donnezan et al[50] | France | MCI | 4 | Physical training | Cognitive training | Simultaneous cognitive and physical training | Control | – | 4 |
| DiNapoli et al[51] | USA | MCI | 2 | Individualised social activities | Control | – | – | – | 4 |
| Doi et al[52] | Japan | MCI | 2 | Exercise | Active control | – | – | – | 4 |
| Doi et al[53] | Japan | MCI | 3 | Dance | Playing musical instruments | Health education group | – | – | 4 |
| Drumond Marra et al[54] | Brazil | MCI | 2 | TMS | Sham TMS | – | – | – | 6 |
| Emsaki et al[55] | Iran | MCI | 2 | Cognitive training | Active control | – | – | – | 9 |
| Eyre et al[56] | USA | MCI | 2 | Yoga | Cognitive training | – | – | – | 10 |
| Feng et al[57] | China | MCI | 2 | Single component cognitive training | Multiple component cognitive training | – | – | – | 3 |
| Fernández-Calvo et al[58] | Spain | Dementia | 2 | Multi-intervention programme | Control | – | – | – | 21 |
| Fiatarone Singh et al[59] | Australia | MCI | 4 | Progressive resistance training and sham cognitive training | Progressive resistance training and cognitive training | Cognitive training | Control | – | 12 |
| Finn and McDonald[60] | Australia | MCI | 2 | Repetition-lag training | Control | – | – | – | 6 |
| Fogarty et al[61] | Canada | MCI | 2 | Memory intervention programme and tai chi | Memory intervention programme | – | – | – | 5 |
| Förster et al[62] | Germany | Both | 2 | Cognitive training | Control | – | – | – | 10 |
| Galante et al[63] | Italy | Dementia | 2 | Cognitive training | Active control | – | – | – | 12 |

Continued

| Study | Country | Diagnosis | Number of groups | Group 1 | Group 2 | Group 3 | Group 4 | Group 5 | Number of measures |
|---|---|---|---|---|---|---|---|---|---|
| Greenaway et al[21] | USA | MCI | 2 | Memory support system (memory rehabilitation) with training | Memory support system without training | – | – | – | 15 |
| Hagovská et al[64] | Czech Republic | MCI | 2 | Cognitive training (computer based) | Cognitive training | – | – | – | 0 |
| Hagovská et al[65] | Czech Republic | MCI | 2 | Cognitive training and dynamic balance training | Balance training | – | – | – | 4 |
| Han et al[66] | South Korea | MCI | 2 | Ubiquitous spaced retrieval-based memory advancement and rehabilitation training | Control | – | – | – | 4 |
| Han et al[67] | South Korea | Both | 2 | Multimodal cognitive enhancement therapy | Active control | – | – | – | 7 |
| Hattori et al[29] | Japan | Dementia | 2 | Art therapy | Active control | – | – | – | 4 |
| Ho et al[68] | Hong Kong | Both | 3 | Dance movement therapy | Physical exercise | Control | – | – | 7 |
| Horie et al[69] | Brazil | MCI | 2 | Group weight loss programme | Control | – | – | – | 10 |
| Hyer et al[70] | USA | MCI | 2 | Cognitive training (computer based) | Active control | – | – | – | 3 |
| Jansen et al[22] | The Netherlands | Dementia | 2 | Case management | Control | – | – | – | 5 |
| Jean et al[71] | Canada | MCI | 2 | Cognitive training | Active control | – | – | – | 10 |
| Jelcic et al[72] | Italy | Dementia | 2 | Lexical-semantic treatment | Cognitive stimulation | – | – | – | 11 |
| Jeong et al[73] | South Korea | MCI | 2 | Cognitive intervention (group based) | Cognitive intervention (home based) | – | – | – | 8 |
| Kinsella et al[23] | Australia | MCI | 2 | Cognitive intervention | Control | – | – | – | 4 |
| Kohanpour et al[74] | Iran | MCI | 4 | Aerobic exercise | Lavender extract | Aerobic exercise and lavender extract | Control | – | 14 |
| Koivisto et al[24] | Finland | Dementia | 2 | Psychosocial intervention | Control | – | – | – | 7 |
| Kovács et al[75] | Hungary | MCI | 2 | Multimodal exercise | Control | – | – | – | 1 |
| Küster et al[76] | Germany | MCI | 3 | Cognitive training | Physical training | Control | – | – | 7 |
| Kwok et al[77] | Hong Kong | MCI | 2 | Cognitive training | Active control | – | – | – | 5 |
| Lam et al[78] | Hong Kong | MCI | 2 | Tai Chi | Stretching | – | – | – | 4 |
| Lam et al[79] | Hong Kong | MCI | 4 | Cognitive training | Cognitive and physical training | Physical training | Social groups | – | 2 |
| Lam et al[25] | Hong Kong | Dementia | 2 | Case management | Control | – | – | – | 2 |
| Langoni et al[80] | Brazil | MCI | 2 | Group exercise | Control | – | – | – | 14 |
| Law et al[81] | Hong Kong | MCI | 2 | Functional tasks exercise programme | Cognitive training | – | – | – | 7 |
| Lazarou et al[82] | Greece | MCI | 2 | Ballroom dancing | Control | – | – | – | 5 |
| Li et al[83] | China | MCI | 2 | Computerised cognitive training | Control | – | – | – | 4 |

Continued

**Table 1** Continued

| Study | Country | Diagnosis | Number of groups | Group 1 | Group 2 | Group 3 | Group 4 | Group 5 | Number of measures |
|---|---|---|---|---|---|---|---|---|---|
| Lim et al[84] | Singapore | MCI | 2 | Mindfulness | Health education | – | – | – | 5 |
| Logsdon et al[26] | USA | Dementia | 2 | Early stage memory loss support group | Control | – | – | – | 10 |
| Luijpen et al[85] | The Netherlands | MCI | 2 | TENS | Sham TENS | – | – | – | 6 |
| Maffei et al[86] | Italy | MCI | 2 | Multidomain training | Control | – | – | – | 10 |
| İnel Manav and Simsek[87] | Turkey | Dementia | 2 | Reminiscence therapy | Social interview | – | – | – | 6 |
| Melendez et al[88] | Spain | Both | 2 | Reminiscence therapy | Control | – | – | – | 6 |
| Nagamatsu et al[89] | Canada | MCI | 2 | Aerobic exercise | Resistance training | – | – | – | 13 |
| Olsen et al[90] | Norway | Both | 2 | Animal-assisted therapy | Control | – | – | – | 9 |
| Pantoni et al[91] | Italy | MCI | 2 | Attention process training | Control | – | – | – | 4 |
| Park and Park[92] | South Korea | MCI | 2 | Cognition-specific computer training | Non-specific computer training | – | – | – | 5 |
| Poinsatte et al[93] | USA | MCI | 2 | Aerobic exercise | Stretching | – | – | – | 3 |
| Pongan et al[94] | France | Dementia | 2 | Choral singing | Painting | – | – | – | 14 |
| Poptsi et al[95] | Greece | MCI | 5 | Paper language tasks | Computer language tasks | Oral language tasks | Active control | Control | 4 |
| Qi et al[96] | China | MCI | 2 | Aerobic exercise | Control | – | – | – | 3 |
| Rapp et al[97] | USA | MCI | 2 | Memory enhancement training (multicomponent) | Control | – | – | – | 9 |
| Rojas et al[98] | Argentina | MCI | 2 | Cognitive intervention | Control | – | – | – | 8 |
| Rozzini et al[99] | Italy | MCI | 2 | Cognitive training and AChEIs | AChEIs | – | – | – | 7 |
| Savulich et al[100] | UK | MCI | 2 | Cognitive training | Control | – | – | – | 9 |
| Scherder et al[101] | The Netherlands | MCI | 3 | Walking | Hand and face exercises | Control | – | – | 11 |
| Shimada et al[102] | Japan | MCI | 2 | Physical and cognitive training | Health education group | – | – | – | 7 |
| Shimizu et al[103] | Japan | MCI | 2 | Movement music therapy | Single training task | – | – | – | 4 |
| Simon et al[104] | Brazil | MCI | 2 | Memory training | Active control | – | – | – | 8 |
| Song et al[105] | China | MCI | 2 | Aerobic exercise | Active control | – | – | – | 4 |
| Suzuki et al[106] | Japan | MCI | 2 | Multicomponent exercise group | Active control | – | – | – | 6 |
| Tappen and Hain[27] | USA | Both | 2 | Cognitive training (home based) | Life story interview | – | – | – | 11 |
| Troyer et al[107] | Canada | MCI | 2 | Multicomponent intervention | Control | – | – | – | 6 |
| Tsai et al[108] | Taiwan | MCI | 3 | Aerobic exercise | Resistance training | Control | – | – | 7 |
| Tsantali et al[109] | Greece | Dementia | 3 | Cognitive training | Cognitive stimulation | Control | – | – | 5 |
| van Uffelen et al[110] | The Netherlands | MCI | 4 | Walking | Placebo activity | Folic acid/ Vitamin b supplements | Placebo pills | – | 3 |

Continued

**Table 1** Continued

| Study | Country | Diagnosis | Number of groups | Group 1 | Group 2 | Group 3 | Group 4 | Group 5 | Number of measures |
|---|---|---|---|---|---|---|---|---|---|
| Waldorff et al[28] | Denmark | Dementia | 2 | Multifaceted counselling, education and support | Control | – | – | – | 2 |
| Wei et al[111] | China | MCI | 2 | Handball training | Control | – | – | – | 8 |
| Yang et al[112] | USA | MCI | 2 | Memory enhancement training | Yoga | – | – | – | 3 |
| Yoon et al[113] | South Korea | MCI | 2 | High-speed power strength training | Low-speed strength training | – | – | – | 5 |
| Young et al[114] | Hong Kong | Dementia | 2 | Support groups | Control | – | – | – | 4 |
| Young et al[115] | Hong Kong | MCI | 2 | Holistic health group | Control | – | – | – | 4 |
| Yun et al[116] | South Korea | MCI | 2 | TDS | Sham TDS | – | – | – | 1 |
| Zhao et al[117] | China | MCI | 2 | Creative expression therapy | Cognitive training | – | – | – | 7 |
| Zhu et al[118] | China | MCI | 2 | Dance | Control | – | – | – | 7 |

MCI, mild cognitive impairment; TDS, transcranial direct current stimulation; TENS, transcutaneous electrical nerve stimulation; TMS, transcranial magnetic stimulation.

a caregiver measure, however since then the use of such measures has declined.

## Use of outcome measures by intervention

Table 3 presents diagnosis and type of intervention by the domains measured. Cognition/memory was the most measured domain across all diagnostic groups, followed by BPSD. The third most common domain for MCI studies was physical performance, whereas caregiver measures were the third most common type of measures used in studies of early dementia.

Cognition/memory was measured in all types of intervention. Measures of BPSD were most common in cognitive training interventions and physical activity interventions, however, they were not used by combined cognitive and physical training interventions. Quality of life was measured by studies of case management, cognitive training, psychosocial interventions, physical activity and support groups.

Caregiver measures were used in five types of interventions: case management, cognitive training and psychosocial interventions; followed by arts-based therapy and support groups.

## Use of outcome measures by country

Table 4 presents the country the research was conducted in by outcome measure domain. Generally, there was not much variability in the domain of outcome measures used by country. Cognition/memory was the domain most frequently measured by all countries, followed by BPSD. The majority of studies were conducted in China (including Hong Kong and Taiwan), these studies focused on cognition/memory, BPSD and biological outcome measures. Other than China, only three other countries included biological measures (Iran, Pakistan and the

USA). The USA had the second largest number of studies included in this review, these studies favoured cognition/memory, BPSD, caregiver measures and quality of life. Out of the 24 countries with studies included in this review, less than half (n=9) included measures of quality of life.

## DISCUSSION

In this study, we used a scoping review to map which outcome measures had been used in trials for non-pharmacological treatments of mild dementia and MCI. We extracted 358 individual outcome measures used in 91 trials, only 22% of which were used more than once. We grouped the outcome measures which had been used more than once and examined differences in their use over time, by diagnostic group, country the research was set in and by the type of intervention they were being used to evaluate. Measures of cognition and BPSDs were the most frequently used across all studies and types of intervention.

Perhaps unsurprisingly, measures of cognition or memory are the most prevalent across all countries, diagnostic groups and types of intervention with the MMSE being the most frequently used outcome measure, despite the ADAS-cog having been validated as the gold-standard measure of cognition.[15 30 31] Measuring cognition is central to measuring the progression of dementia and is a clinically and empirically useful outcome to measure in dementia research.[31] However, in this review, we charted 40 different measures of cognition. This indicates that while cognition has been prioritised as an outcome in studies of non-pharmacological interventions, there is no consensus between researchers on which specific

**Table 2** Outcome measures by domain and subdomains

| Person living with dementia measures Domain and subdomain | Outcome measure | N |
|---|---|---|
| **Cognition/Memory** | | **219** |
| Cognition | MMSE | 37 |
| | Trail Making Test | 27 |
| | Digit Span Test | 12 |
| | ADAS-Cog | 10 |
| | Rey Auditory Test | 9 |
| | Rivermead Behavioural Memory Test | 9 |
| | Stroop Test | 7 |
| | MMQ | 7 |
| | Novelli Lexical Test | 7 |
| | MoCA | 6 |
| | CDR | 6 |
| | Verbal Fluency | 6 |
| | CERAD-NB | 5 |
| | Addenbrooke's Cognitive Examination | 4 |
| | Boston Naming Test | 4 |
| | Rey Osterrieth Complex Figure Task | 4 |
| | Montreal Cognitive Test | 3 |
| | Attentional Matrices Test | 3 |
| | California Verbal Learning Test | 3 |
| | Digit Symbol Coding Test | 3 |
| | Hopkins Verbal Learning Test | 3 |
| | The Wechsler Memory Scale | 3 |
| | CAMcog | 2 |
| | Cognitive Failures Test | 2 |
| | Colour Trails Test | 2 |
| | Dementia Rating Scale-2 | 2 |
| | DSM IV Test | 2 |
| | Auditory Verbal Learning Test | 2 |
| | Corsi's Block Tapping Test | 2 |
| | Frontal Assessment Test | 2 |
| | Fuld Object Memory Evaluation | 2 |
| | Logical Memory (Subtest of Wechsler Memory Scale) | 2 |
| | Prospective and Retrospective Memory Questionnaire | 2 |
| | Pyramids & Palm Trees | 2 |
| | Questionnaire d'Auto Evaluation de la Memoire | 2 |
| | Raven's Coloured Matrices | 2 |
| | Repeatable Battery Test | 2 |
| | The verbal learning and memory test | 2 |
| | Visual Memory Span | 2 |

Continued

**Table 2** Continued

| Person living with dementia measures Domain and subdomain | Outcome measure | N |
|---|---|---|
| | Wechsler Adult Intelligence Scale | 2 |
| Knowledge of memory strategies | Memory Strategy Toolbox | 2 |
| | Strategy Knowledge Repertoire | 1 |
| Attention | Test of Everyday Attention | 2 |
| **Behavioural and psychological symptoms of dementia** | | **51** |
| Anxiety/Depression | Geriatric Depression Scale* | 21 |
| | Cornell Scale for Depression in Dementia* | 7 |
| | Hospital Anxiety and Depression Scale | 4 |
| | Beck Depression Inventory | 1 |
| Other | Neuropsychiatric Inventory* | 12 |
| | Apathy Evaluation Scale | 3 |
| | Revised memory and behaviour problem checklist* | |
| **Everyday living** | | **20** |
| Activities of daily living | Instrumental Activities of Daily Living* | 8 |
| | Bayer Activities of Daily Living Scale | 3 |
| | Alzheimer's Disease Cooperative Study Activities of Daily Living Scale | 2 |
| | Barthel Index | 2 |
| Functional ability | Functional Activities Questionnaire | 3 |
| | Functional and Cognitive Assessment Test and Functional Rating Scale for Dementia | 2 |
| **Physical outcomes** | | **19** |
| Physical performance | Timed Up and Go Test | 7 |
| | Gait | 3 |
| | Handgrip strength | 3 |
| | Stride | 2 |
| | Walking Speed | 2 |
| Physical measures | Weight | 2 |
| **Quality of life/Well-being** | | **15** |
| Quality of life | QoL in Alzheimer's disease* | 7 |
| | Dementia Quality of Life Instrument* | 3 |
| | EuroQoL EQ 5D* | 2 |
| | EQ-VAS | 1 |
| Stress | Perceived Stress Scale | 1 |
| General Well-being | SF-36 | 1 |
| **Biological outcome** | | **9** |
| Brain activity | EEG | 4 |
| | MRI | 2 |
| Biomarker | BDNF | 3 |

Continued

| Table 2 | Continued | |
|---|---|---|
| **Person living with dementia measures Domain and subdomain** | **Outcome measure** | **N** |
| **Adherence to intervention** | | **2** |
| Adherence to intervention | Adherence | 2 |
| **Caregiver measures domain** | **Outcome measure** | **N** |
| **Depression** | | **5** |
| | The Center for Epidemiological Studies Depression Scale* | 3 |
| | Geriatric Depression Scale | 1 |
| | Beck Depression Inventory | 1 |
| **Caregiver burden** | | **2** |
| | Zarit caregiver burden interview* | 2 |
| **General well-being** | | **1** |
| | SF-36* | 1 |
| | | |
| **Knowledge of memory strategies** | | **1** |
| | Strategy Knowledge Repertoire | 1 |
| **Quality of life** | | **1** |
| | EQ-VAS | 1 |
| **Stress** | | **1** |
| | Perceived Stress Scale | 1 |

*Measure recommended by INTERDEM Consensus.[14]
CDR, Clinical Dementia Rating; CERAD-NB, Consortium to Establish a Registry for Alzheimer's Disease- Neuropsychological Battery; DSM, Diagnostic Statistical Manual of Mental Disorders; EEG, electroencephalogram; EQ-VAS, EuroQoL Visual Analogue Scales; EuroQoL EQ 5D, EuroQoL 5-dimension; MMQ, Multifactorial Memory Questionnaire; MMSE, Mini-Mental State Examination; MoCA, Montreal Cognitive Assessment; SF-36, 36-Item Short Form Survey.

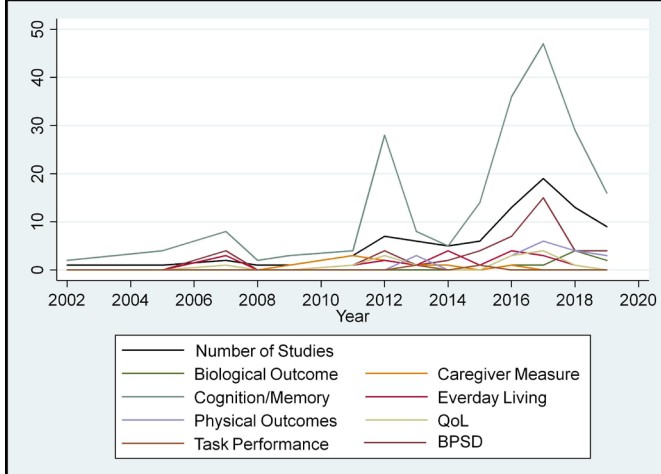

**Figure 2** Trends in outcome measures over time. BPSD, behavioural and psychological symptoms of dementia; QoL, quality of life.

measures should be used. In addition to measures of cognitive performance, three studies have also measured participant's knowledge or retention of memory strategies, indicating an interest in long-term coping strategies for memory loss.

Measures of the BPSD have become more common over time, becoming in 2017 the most measured outcome after cognition. There is not much variety in the BPSDs which have been measured. Generally, depression was measured over other BPSDs. Other BPSDs such as agitation were measured less, perhaps because they are more associated with the later stages of the disease and depression is associated with the earlier stages.[32]

Quality of life and well-being were not among the most measured domains. Four measures of quality of life were used 15 times across the included studies and all but one of these measures were dementia-specific measures. It is surprising quality of life has not been measured more, as previous research has stated that in the absence of a cure, healthcare providers have a greater ability to improve quality of life than alter the progression of the disease.[33] Furthermore, both people with MCI and caregivers rated quality of life of the patient as the most important outcome to measure, followed by caregiver quality of life/burden.[34] Indicating while quality of life has been identified as a priority by PLWD, people diagnosed with MCI and their caregivers in previous research, the findings of this study shows this is not being translated into trials of non-pharmacological treatments for early dementia and MCI.

Likewise, caregiver measures had consistent low use across the studies included in this review. We charted eight caregiver measures which were used 11 times across the included studies. Caregiver measures were more commonly used in studies of PLWD, rather than MCI. Previous research has highlighted the profound effect of dementia on their caregivers, with around half of caregivers experiencing high levels of burden.[35] However, a third of caregivers of people with MCI also report extreme levels of burden,[36] yet the findings of this study show this is less investigated.

There was great variability in the types of outcomes being used to evaluate the different types of intervention. All studies measured cognition and all but one measured BPSD. A lack of clarity in how change occurs as a result of non-pharmacological treatments is a fundamental weakness in this area of work.[4] It is unlikely that all interventions being tested in this review could hope to improve cognition, however this is the most prevalent domain of outcome measures. There are a number of practical reasons as to why certain outcomes, and therefore outcome measures are used over others, In the past, pharmacological treatments have been required to include some measure of cognition, functional or global assessment,[17] it is possible that this approach has influenced the choice in outcomes used in non-pharmacological studies. Furthermore, some measures may be used over others for more practical reasons. For example, measures

**Table 3** Outcome measure domain by diagnosis and intervention

| | Number of studies | BPSD | Biological outcome | Caregiver measure | Cognition/Memory | Everyday living | Physical measures | Physical performance | Quality of life/ Well- being | Task performance |
|---|---|---|---|---|---|---|---|---|---|---|
| **Diagnosis** | | | | | | | | | | |
| Both | 6 | 5 | – | 1 | 12 | 1 | – | – | – | – |
| Dementia | 14 | 16 | – | 7 | 42 | 6 | – | – | 6 | – |
| MCI | 71 | 30 | 9 | 3 | 163 | 12 | 2 | 17 | 9 | 2 |
| **Type of intervention** | | | | | | | | | | |
| Animal-assisted therapy | 2 | 2 | – | – | 2 | 1 | – | – | – | – |
| Art-based therapy | 2 | 1 | – | 1 | 6 | 1 | – | – | – | – |
| Case management | 2 | 2 | – | 3 | 1 | – | – | – | 1 | – |
| Chinese calligraphy | 2 | 1 | 1 | – | 4 | – | – | – | – | – |
| Cognitive training | 37 | 23 | 2 | 3 | 103 | 11 | – | 1 | 6 | 2 |
| Cognitive training and physical activity | 4 | – | – | – | 14 | 2 | – | 2 | – | – |
| Multicomponent psychosocial intervention | 4 | 6 | – | 3 | 10 | 2 | – | 2 | 3 | – |
| Music-based intervention | 2 | 1 | – | – | 7 | – | 1 | 2 | 1 | – |
| Physical activity | 25 | 11 | 6 | – | 53 | 3 | 1 | 10 | 2 | – |
| Reminiscence therapy | 2 | 1 | – | – | 2 | – | – | – | – | – |
| Support group | 3 | 3 | – | 1 | 1 | – | – | – | 1 | – |

BPSD, behavioural and psychological symptoms of dementia; MCI, mild cognitive impairment.

which are short to administer and free to use may be priorities over others.[31] Several interventions in this review comprise more than one component, for example, physical activity and cognitive training. In these cases, it may take multiple measures over many domains to accurately capture change. It is vital that outcome measures are selected depending on the domains the intervention is seeking to address.[31]

In 2008, the INTERDEM group recommended 22 outcome measures for use across 9 domains.[15] We found 11 of these 22 measures (50%) were used by the studies included in this review, one of the recommended domains (staff carer morale) was not applicable to the studies included in this review. All measures recommended for measuring patient mood, and patient quality of life were charted in this review. Only one of the recommended measures for the activities of daily living, caregiver mood, caregiver burden and caregiver quality of life domains were charted and no measures under the global measures domain were charted in this review. This indicates that there is some consistency between which measures are recommended and which measures are used, this is largely for patient measures and there is less consistency for caregiver measures.

In this study, we found that the use of outcome measures did not vary much by the country the study was conducted in. In each country, cognition/memory was the most commonly tested domain, followed by BPSD. The importance of outcomes may vary between cultures; therefore, it is important that the outcomes and measures used reflect this.[16] However, due to the limitations of the methodology used we cannot comment on the cultural relevance of the outcome measures charted in this review. Furthermore, articles were only included if they were published in English. It is possible that more culturally appropriate outcomes were used in articles published in the same language as the population under investigation. This is an important area for future research.

## Limitations

The findings of this review must be interpreted in the context of the study. To make this review feasible we only included full RCTs, other outcome measures may have been used in different types of studies. Due to time constraints, some subtypes of dementia and cognitive impairment (young-onset, Parkinson's disease dementia and vascular cognitive impairment) were excluded from this review, which limits the applicability of these findings. Further research is needed to explore whether the pattern in the use of outcomes and outcome measures is similar in these groups, compared with the ones included in this review. Furthermore, only outcome measures which were

**Table 4** Outcome measure domain by country

| Country | Number of studies | BPSD | Biological outcome | Caregiver measure | Cognition/Memory | Functional ability | Physical measures | Physical performance | Quality of life/Well-being | Task performance |
|---|---|---|---|---|---|---|---|---|---|---|
| Argentina | 1 | 1 | 0 | 0 | 6 | 1 | 0 | 0 | 0 | 0 |
| Australia | 4 | 0 | 0 | 1 | 5 | 1 | 0 | 0 | 0 | 0 |
| Brazil | 5 | 1 | 1 | 0 | 14 | 0 | 0 | 1 | 0 | 0 |
| Canada | 6 | 2 | 0 | 0 | 16 | 0 | 0 | 2 | 0 | 0 |
| Mainland China, Hong Kong and Taiwan | 20 | 10 | 5 | 1 | 35 | 2 | 0 | 0 | 0 | 1 |
| Czech Republic | 3 | 0 | 0 | 0 | 3 | 2 | 0 | 1 | 0 | 0 |
| Denmark | 1 | 2 | 0 | 2 | 1 | 1 | 0 | 0 | 2 | 0 |
| Finland | 1 | 1 | 0 | 1 | 3 | 1 | 0 | 0 | 1 | 0 |
| France | 3 | 1 | 0 | 0 | 6 | 0 | 0 | 2 | 1 | 0 |
| Germany | 4 | 1 | 0 | 0 | 10 | 0 | 0 | 0 | 1 | 0 |
| Greece | 4 | 3 | 0 | 0 | 18 | 2 | 0 | 0 | 1 | 0 |
| Hungary | 1 | 0 | 0 | 0 | 0 | 0 | 0 | 1 | 0 | 0 |
| Iran | 3 | 1 | 1 | 0 | 3 | 0 | 1 | 0 | 0 | 0 |
| Italy | 11 | 8 | 0 | 0 | 32 | 6 | 0 | 0 | 1 | 0 |
| Japan | 8 | 2 | 0 | 1 | 16 | 1 | 1 | 6 | 0 | 0 |
| Norway | 1 | 1 | 0 | 0 | 1 | 0 | 0 | 0 | 0 | 0 |
| Pakistan | 1 | 0 | 1 | 0 | 3 | 0 | 0 | 0 | 0 | 0 |
| Singapore | 1 | 0 | 0 | 0 | 0 | 0 | 0 | 0 | 0 | 0 |
| South Korea | 8 | 5 | 0 | 0 | 14 | 1 | 0 | 4 | 3 | 0 |
| Spain | 3 | 2 | 0 | 0 | 2 | 0 | 0 | 0 | 0 | 0 |
| The Netherlands | 5 | 0 | 0 | 2 | 10 | 0 | 0 | 0 | 2 | 0 |
| Turkey | 1 | 1 | 0 | 0 | 1 | 0 | 0 | 0 | 0 | 0 |
| UK | 1 | 3 | 0 | 0 | 1 | 0 | 0 | 0 | 0 | 0 |
| USA | 10 | 6 | 1 | 3 | 19 | 2 | 0 | 0 | 3 | 1 |

BPSD, behavioural and psychological symptoms of dementia.

published could be included in this review. The studies included in this study were heterogeneous in terms of participants recruited, interventions tested and outcome measures used, making it difficult to group them thematically. It is possible some nuance is lost in the exploration of broader themes. As with the nature of scoping reviews, we are only able to present which outcome measures have been used in previous research, we are unable to draw conclusions as to which outcome measures should be used over others. Future research should explore which population measures have been validated for and what constitutes a clinically useful change.

### Implications and recommendations for future research

The findings of this review indicate there is very little consistency in outcome measures used in RCTs for non-pharmacological interventions in MCI and mild dementia, however we are not able to conclude which measures should be used over others. To create a strong evidence base for non-pharmacological treatments more

research, with the involvement of PLWD and their carers, is needed to determine which measures are preferable over a greater number of domains. Additionally, the prevalence of cognitive measures found in this study suggests that researchers are including such measures because there is an expectation to do so. Researchers should be clear on the theory behind how their intervention creates change and use the appropriate outcome measures.

### CONCLUSIONS

In summary, this study has found RCTs for non-pharmacological treatments in mild dementia and MCI use a broad range of outcome measures, with a small proportion being used more than once. Excepting measures of cognition, there is very little commonality between studies. Where previous research has set priorities on outcomes preferred by PLWD, people with MCI and caregivers, quality of life, for example, this has not yet

translated into studies measuring new treatments. Further research is needed to understand which outcomes should be prioritised and how they should be measured.

**Contributors** EC designed the study, carried out the literature review, the data charting and synthesis, data interpretation, article preparation, article review and correspondence. MP and VL contributed to the study design, data interpretation and article review. MC contributed to the data charting.

**Funding** EC is supported by a studentship from the ESRC LISS-DTP.

**Competing interests** None declared.

**Patient and public involvement** Patients and/or the public were involved in the design, conduct, reporting, dissemination plans of this research. Refer to the 'Methods' section for further details.

**Patient consent for publication** Not required.

**Provenance and peer review** Not commissioned; externally peer reviewed.

**Data availability statement** No data are available.

**ORCID iD**
Elyse Couch http://orcid.org/0000-0003-4692-5837

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
