## [Reviewer comments · BMJ Open]

ARTICLE DETAILS

TITLE (PROVISIONAL)	Outcomes tested in non-pharmacological interventions in mild cognitive impairment and mild dementia: a scoping review
AUTHORS	Couch, Elyse; Lawrence, Vanessa; Co, Melissa; Prina, A. Matthew

VERSION 1 – REVIEW

REVIEWER	Dr Jenny McCleery Oxford Health NHS Foundation Trust UK
REVIEW RETURNED	04-Dec-2019

GENERAL COMMENTS	The issue of diversity of outcome measures in dementia research is an important one and this paper contributes to the delineation of the problem. The authors should refer to previous publications about similar work; I am not sure it is accurate to claim this is the first review to explore trends in this area. I have some concerns about lack of clarity of the exact scope of the work related to terminology and, more importantly, to the inclusion/exclusion criteria. The paper is generally well-written. The methods are clearly described and the tables are helpful. Introduction The authors refer to the work conducted by INTERDEM in 2008, which concentrated on recommendations for European research. They state that it is important to have a more coherent use of outcome measures internationally and, indeed, to seek global consistency in outcome measures. This is by no means self-evident. It seems perfectly plausible that different cultures and care systems may have different priorities and different values (e.g. related to key aspects of QoL) and that local decisions about key outcome measures may have advantages, perhaps more for some types of intervention than others (e.g. case management or similar). The authors fail to refer to some important prior publications on this topic. For example, the International Psychogeriatric Association in 2007 published a statement on outcomes in dementia research: a recent systematic scoping review, similar to this work, which also looked at time trends: Harrison, J.K., Noel-Storr, A.H., Demeyere, N. et al. Outcomes measures in a decade of dementia and mild cognitive impairment trials. Alz Res Therapy 8, 48 (2016) doi:10.1186/s13195-016-0216-8. The authors should also refer to a large project with which this work overlaps extensively, and which extends the work presented here: Harding, A.J.E., Morbey, H., Ahmed, F. et al. Developing a core outcome set for people living with dementia at home in their
--

neighbourhoods and communities: study protocol for use in the evaluation of non-pharmacological community-based health and social care interventions. *Trials* 19, 247 (2018)
doi:10.1186/s13063-018-2584-9

Study question and methods

Use of the term 'early dementia' is confusing. Patients and carers are often uncertain whether this means 'early-onset' or 'early in the course of the illness'. Even 'early-stage dementia' could refer to time since diagnosis or to severity of symptoms? Do the authors mean 'mild dementia'? The inclusion criteria should explain how this was defined.

It is not clear why the authors chose only to review studies of non-pharmacological interventions. There is no obvious reason why patients should want different outcomes from drug and non-drug interventions or why the best measures to assess these outcomes would vary with the nature of the intervention. This decision to limit the scope of the review should be justified.

Similarly, it is not clear why studies of people with vascular cognitive impairments should be excluded. Many studies of non-pharmacological interventions and many MCI studies in particular do not try make precise subtype diagnoses (and indeed the authors include studies which identify participants using simple clinical scales such as MMSE), so was it in fact possible to implement this exclusion criterion reliably? Why should key outcomes and the choice of outcome measure differ because the illness trajectory may be somewhat different? Indeed, this begs the question 'different from what'? Is the review intended to cover only mild dementia due to probable AD and probable AD-MCI? If so, this should be stated. Were studies of people with DLB, PDD and rarer dementias included or excluded? There are RCTs on, for example, cognitive training/rehab in people with MCI in PD which would appear to meet the inclusion criteria, but do not appear to have been included (e.g. Costa et al. Prospective memory performance of patients with Parkinson's disease depends on shifting aptitude: evidence from cognitive rehabilitation. *J Int Neuropsychol Soc* 2014; 20:717-26; Lawrence et al. Cognitive Training and Transcranial Direct Current Stimulation for Mild Cognitive Impairment in Parkinson's Disease: A Randomized Controlled Trial. *Parkinson's Disease* 2018; Article ID 4318475; Petrelli et al. Effects of cognitive training in Parkinson's disease: A randomized controlled trial. *Parkinsonism and Related Disorders* 2014; 20:1196-1202; and others).

Lastly on the exclusion criteria, the authors should justify excluding studies intended to treat depression. Do they mean major depressive disorder or depressed mood / depressive symptoms? If the latter, then what is the reason for treating studies targeting this symptom domain differently from those targeting other non-cognitive symptoms?

Additional points

A small point, but the title of Table 3 is incorrect. As the text correctly explains, this tabulates outcome domains (not outcome measures) against type of intervention. (The distinction matters because the authors have argued in the Introduction that

	researchers should be careful to match the outcomes (i.e. outcome domains) they assess in their studies to the nature of the intervention). The authors' Conclusions also confuse outcomes and outcome measures in the third and fourth sentences. The utility of the review would be greatly enhanced by additional information on the outcome measures. For example, which outcome measures have been validated in the population of interest? For which outcome measures is there any published information on a minimum clinically important difference? (MCID of outcome measures is a much-neglected aspect of outcome measures which is essential for interpreting research findings). Not including these features is to some extent a wasted opportunity to add to the literature in an important way and to influence the research agenda.
--	---

REVIEWER	Dr. Garuth Chalfont Research Associate C4AR - Centre for Ageing Research Division of Health Research Faculty of Health and Medicine Furness College – Room C83 Lancaster University Lancaster LA1 4YG United Kingdom
REVIEW RETURNED	23-Dec-2019

GENERAL COMMENTS	Review of Couch et al. Outcomes tested in non-pharmacological interventions in MCI and early dementia: a scoping review (2020, BMJ Open)		
	#	Checklist	
		Page, line	Comment Done
	1	12, 5	(See note #2)
	2	7, 17	'care homes' does not have an agreed international meaning - some consider nursing homes as care homes; Japan uses the term 'group homes'... maybe long term care or an institutional setting or facility.... Not sure which to recommend, but just wanted to raise the point about universal nomenclature.
	3	7, 19	Some may question why you exclude papers not in English if you are aiming to look at outcome measures globally... obviously if you are funded to go deeper you would want to include these.

4	7, 27	Not sure about exclusion: 'vascular cognitive impairment or young onset' Did you have any that were excluded on this point? What does a different trajectory of decline have to do with measuring outcomes? (No action needed, just wondering)	
5	13, 16	This is a good point. (No action needed, just saying)	
6	14, 13	"Outcome measures should be selected depending on the domains the intervention is seeking to address ²⁸ " (see note #6 below)	
	Page, line	Typos, grammar, clarity, etc....	Done
	2, 41	'RESULTS' is in caps ?	
	4, 17	Insert a period after "treatments ⁴ "	
	4, 45	fix "...not recommended, however there is..."	
	7, 3	fix "...at both at..."	
	7, 12	Break into 2 sentences "...included, however,"	
	7, 38	fix "...measure were..."	
	7, 47	new sentence "February 2018, an additional"	
	8, 45	fix... "outcome measures which used more than once"	
	8, 48	new sentence "domain, we grouped"	
	9, 26	fix "number of outcomes measured used"	
	9, 40	add to "these countries had fewer than 5 included studies each."	
	9, 48	fix "Results of indivial sources of evidence"	
	11, 43	a bit awkward... "outcome measures domains"	
	11, 52	"until around 2012, where it overtakes" should it be 'when' ?	

	11, 55	new sentence... “everyday living, however,”	
	11, 55	“the number of studies including these measures have declined.” should bit e ‘number has declined’ ?	
	12, 16	“...common domain measured was caregiver specific measures...” is unclear	
	12, 21-25	Clarify meaning or break into 2 clear sentences “BPSD was.... psychosocial interventions.”	
	13, 3	Add a word in “...and is a clinically and empirically useful outcome...”	
	13, 6	Start new sentence “...however, in this review,...”	
	13, 31	fix verb “quality of life and wellbeing was not amongst...”	
	13, 33	Start new sentence “...across the included studies, all but one...”	
	13, 41	“priority setting exercise” is a bit unclear until you get further into the sentence. Either hyphenate ‘priority-setting’ or change sentence around/shorten it	
	13, 47	Sentence may be clearer if you replace ‘Indicating’ with ‘Therefore,’ and add ‘or’ between “PLWD, MCI”	
	14, 25	fix “...patient quality of life and patient quality of life...”	
	15, 8	fix “...however we care not able to...”	
	15, 15	fix “..which measures preferable over.”	
	15, 39	fix this sentence... “Further research to understand which outcomes should be prioritised and how they should be measured.”	
	Ref #115	Capitalize ‘effects’	

Notes:

1. Excellent topic, thank you for your interest in this area.
2. You may want to mention the inclusion of (or lack of) theoretical underpinning to a particular intervention. Line 12, page 5 “It is important researchers are clear on which domains their interventions are targeting, ...” alludes to this weakness. Theoretical underpinning for the intervention (if given at all) may indicate whether it is enacted as a treatment for an underlying driver of the disease, to slow progression and/or actually reverse the disease course, or whether it is enacted solely to modify symptoms. Very few interventions are theoretically grounded and evidence-based to address the underlying course of the disease. (on page 15, line 20 you do mention ‘theory’ so you have been thinking about it, just make more of it.)
3. (Why so few studies in the UK...? No response required... just wondering.....)
4. Your comment is true but only a part of the picture. Sheehan (2012) gives good comparisons between various tools, raising points like the amount of time needed to administer, the expertise required of the person using it, and the availability of the tool in various languages. “it should be practical to use – in practice, this often depends on it being short (so it can be used in busy clinical practice or as an outcome measure in a trial such that participants are not overburdened by long interviews” The topic of outcome measurement tools is separate from which outcomes to measure, but is tied to it to some degree. The decision to measure a certain outcome is driven by perhaps a number of underlying issues. You may need to just add a paragraph in the discussion section about this so readers are not left with the impression that outcomes are being recommended with no understanding of the methodological implications. Did the studies shed any light at all on why certain tools were chosen over others...? I doubt it, but there may be a paper or two on this topic - not of which outcomes are recommended, but of which tools are used and why. Some tools are available on the web for free and others are under license. Economics and resource issues probably drive the use of some tools over others. The MMSE might be the McDonald's of tools..... Cheap (or free), quick and available everywhere...!
5. You may want to mention the preponderance of multimodal interventions. These are important as cognitive impairment and dementia are recognized as multifactorial syndromes, so it takes multiple domains or modes to address it holistically. Various terms found in your reviewed papers were: multimodal, multidomain, multi-intervention, cognitive training and dynamic balance, memory enhancement training, physical and cognitive training, multicomponent exercise, multicomponent intervention and holistic health group. There were an average of 6.9 measures per study.

	Lead author	Reference #	# of measures
1	Bae	35	10
2	Combourieu & Donnezan	47	4
3	Feng	54	3
4	Fernandez-Calvo	55	21
5	Fiatarone Singh	56	12
6	Fogarty	58	5
7	Hagovska	62	4
8	Han	64	7
9	Kovacs	72	1
10	Lam	76	2
11	Maffei	83	10
12	Rapp	94	9
13	Shimada	99	7
14	Suzuki	103	6
15	Troyer	104	6
16	Young	113	4
		TOTAL	111

6. You may want to add this recent reference for INTERDEM: Vernooij-Dassen, M., et al. (2019). "Bridging the divide between biomedical and psychosocial approaches in dementia research: the 2019 INTERDEM manifesto." *Aging Ment Health*: 1-7. 10.1080/13607863.2019.1693968

Or this one from the JPND which has developed outcome measures:

Oksnebjerg, L., et al. (2018). "Towards capturing meaningful outcomes for people with dementia in psychosocial intervention research: A pan-European consultation." *Health Expect* 21(6): 1056-1065. 10.1111/hex.12799

I'm not sure if these are the same team or not....?? They have some of the same experts on them.

7. If your aim in the next phase of this work is that "more research is needed to highlight which outcome measures

	should be used over others” then perhaps add a sentence about how you might decide that. In other words, what do you see wrong with the ones in use.... Is it all about facilitating meta-analyses or is there more to it than that...? 8. I dug through my Endnote and thought these studies may be of interest down the road.... Bademli, K., et al. (2018). "Effects of Physical Activity Program on cognitive function and sleep quality in elderly with mild cognitive impairment: A randomized controlled trial." Perspect Psychiatr Care. 10.1111/ppc.12324 Barban, F., et al. (2016). "Protecting cognition from aging and Alzheimer's disease: a computerized cognitive training combined with reminiscence therapy." International Journal Of Geriatric Psychiatry 31(4): 340-348. 10.1002/gps.4328 de Oliveira Silva, F., et al. (2019). "Three months of multimodal training contributes to mobility and executive function in elderly individuals with mild cognitive impairment, but not in those with Alzheimer's disease: A randomized controlled trial." Maturitas 126: 28-33. 10.1016/j.maturitas.2019.04.217 Fortier, M., et al. (2019). "A ketogenic drink improves brain energy and some measures of cognition in mild cognitive impairment." Alzheimers Dement 15(5): 625-634. 10.1016/j.jalz.2018.12.017 Iyalomhe, O., et al. (2015). "A standardized randomized 6-month aerobic exercise-training down-regulated pro-inflammatory genes, but up-regulated anti-inflammatory, neuron survival and axon growth-related genes." Exp Gerontol 69: 159-169. 10.1016/j.exger.2015.05.005 Jiang, Y. and S. Sun (2019). "Blueberry Extracts Supplementation Improves Cognitive Function in Elderly Patients with MCI." Neuroscience, Cognitive Function and Chronobiology. Park, H., et al. (2019). "Combined Intervention of Physical Activity, Aerobic Exercise, and Cognitive Exercise Intervention to Prevent Cognitive Decline for Patients with Mild Cognitive Impairment: A Randomized Controlled Clinical Study." J Clin Med 8(7). 10.3390/jcm8070940 Regan, B., et al. (2017). "MAXCOG-Maximizing Cognition: A Randomized Controlled Trial of the Efficacy of Goal-Oriented Cognitive Rehabilitation for People with Mild Cognitive Impairment and Early Alzheimer Disease." Am J Geriatr Psychiatry 25(3): 258-269. 10.1016/j.jagp.2016.11.008
--	--

	Rovner, B. W., et al. (2018). "Preventing Cognitive Decline in Black Individuals With Mild Cognitive Impairment: A Randomized Clinical Trial." JAMA Neurol 75(12): 1487-1493. 10.1001/jamaneurol.2018.2513 Tao, J., et al. (2019). "Mind-body exercise improves cognitive function and modulates the function and structure of the hippocampus and anterior cingulate cortex in patients with mild cognitive impairment." Neuroimage Clin 23: 101834. 10.1016/j.nicl.2019.101834 Teixeira, C. V. L., et al. (2018). "Cognitive and structural cerebral changes in amnesic mild cognitive impairment due to Alzheimer's disease after multicomponent training." Alzheimer's & Dementia: Translational Research & Clinical Interventions. 10.1016/j.trci.2018.02.003 Reviews of interest.....? Harrison, J. K., et al. (2016). "Outcomes measures in a decade of dementia and mild cognitive impairment trials." Alzheimers Res Ther 8(1): 48. 10.1186/s13195-016-0216-8 Horr, T., et al. (2015). "Systematic review of strengths and limitations of randomized controlled trials for non-pharmacological interventions in mild cognitive impairment: Focus on Alzheimer's disease." J Nutr Health Aging 19(2): 141-153. Karssemeijer, E. G. A., et al. (2017). "Positive effects of combined cognitive and physical exercise training on cognitive function in older adults with mild cognitive impairment or dementia: A meta-analysis." Ageing Res Rev 40: 75-83 10.1016/j.arr.2017.09.003 Ozbe, D., et al. (2019). "Immediate Intervention Effects of Standardized Multicomponent Group Interventions on People with Cognitive Impairment: A Systematic Review." J Alzheimers Dis 67(2): 653-670. 10.3233/JAD-180980 9. Charts and tables are very clear and well-presented...!
--	---

VERSION 1 – AUTHOR RESPONSE

Reviewer 1

Comment 1: The authors refer to the work conducted by INTERDEM in 2008, which concentrated on recommendations for European research. They state that it is important to have a more coherent use of outcome measures internationally and, indeed, to seek global consistency in outcome measures. This is by no means self-evident. It seems perfectly plausible that different cultures and care systems may have different priorities and different values (e.g. related to key aspects of QoL) and that local decisions about key outcome measures may have advantages, perhaps more for some types of intervention than others (e.g. case management or similar).

Response 1: We thank the reviewer for this insightful and important comment. We agree that different cultures may have different priorities for outcome measures, which is supported by previous reviews in this area. We felt that this was so important that we have amended the statements calling for more global consistency, and instead made the case that the use of outcome measures needs to be more culturally sensitive. We have added it as an aim to this study to explore the domain of outcome measure used by the country the research was conducted in.

Objective 3 (page 6) now reads:

“(3) Explore whether the outcome measures used differ depending on the type of intervention, study population, and country the research was conducted in.”

Page 5 lines 22-25 now reads:

“The outcome measures were selected based on their applicability to European research. The utility of outcome measures may vary by culture¹⁶, previous reviews exploring the use of outcome measures in dementia research have not investigated how this differs by country¹⁷.”

We have produced a new table (Table 4, page 35) to present the results from this analysis. In the results section, we have added an analysis of the use of outcome measures by the country the research was conducted in (P13 lines 10-19):

Use of outcome measures by country

“Table 4 presents the country the research was conducted in by outcome measure domain. Generally, there was not too much variability in the domain of outcome measures used by country. Cognition/memory was the domain most frequently measured by all countries, followed by BPSD. The majority of studies were conducted in China (including Hong Kong and Taiwan), these studies focused on cognition/memory, BPSD and biological outcome measures. Other than China, only three other countries included biological measures (Iran, Pakistan and the USA). The USA had the second largest number of studies included in this review, these studies favoured cognition/memory, BPSD, caregiver measures and quality of life. Out of the 24 countries with studies included in this review, less than half (n=9) included measures of quality of life.”

We have also considered these results in the discussion (P16 lines 8-15):

“In this study, we found that the use of outcome measures did not vary much by the country the study was conducted in. In each country, cognition/memory was the most commonly tested domain, followed by BPSD. The importance of outcomes may vary between cultures; therefore, it is important that the outcomes and measures used reflect this ¹⁶. However, due to the limitations of the methodology used we cannot comment on the cultural relevance of the outcome measures charted in this review. Furthermore, articles were only included if they were published in English. It is possible

that more culturally appropriate outcomes were used in articles published in the same language as the population under investigation. This is an important area for future research.”

Comment 2: The authors fail to refer to some important prior publications on this topic. For example, the International Psychogeriatric Association in 2007 published a statement on outcomes in dementia research: a recent systematic scoping review, similar to this work, which also looked at time trends: Harrison, J.K., Noel-Storr, A.H., Demeyere, N. et al. Outcomes measures in a decade of dementia and mild cognitive impairment trials. *Alz Res Therapy* 8, 48 (2016) doi:10.1186/s13195-016-0216-8. The authors should also refer to a large project with which this work overlaps extensively, and which extends the work presented here: Harding, A.J.E., Morbey, H., Ahmed, F. et al. Developing a core outcome set for people living with dementia at home in their neighbourhoods and communities: study protocol for use in the evaluation of non-pharmacological community-based health and social care interventions. *Trials* 19, 247 (2018) doi:10.1186/s13063-018-2584-9

Response 2: We thank the reviewer for drawing attention to these very relevant publications. We have included citations for these studies. They were particularly helpful for addressing the issues raised in comment 1.

Comment 3: Use of the term ‘early dementia’ is confusing. Patients and carers are often uncertain whether this means ‘early-onset’ or ‘early in the course of the illness’. Even ‘early-stage dementia’ could refer to time since diagnosis or to severity of symptoms? Do the authors mean ‘mild dementia’? The inclusion criteria should explain how this was defined.

Response 3: We agree that the term early-dementia can be confusing, and that mild dementia better describes the stage of the disease we are exploring. Therefore, we have amended any use of “early stage dementia” to be mild dementia- including in the title and abstract.

We agree that the inclusion criteria should include more detail as to how this is defined. The inclusion criteria for this point now reads (Page 7, lines 4-17):

- “Participants had a diagnosis of MCI or mild dementia, which was either diagnosed in clinical practice, or met standardised diagnostic criteria, such as the International Statistical Classification of Diseases (ICD-10) or The Diagnostic Statistical Manual of Mental Disorders (DSM), The National Institute of Communicative disorders and Stroke and the Alzheimer’s Disease and Related Disorders (NINCDS-ADRDA), the International working group on MCI criteria, The Consortium to Establish a Registry for Alzheimer’s Disease (CERAD), The National Institute on Aging- Alzheimer’s Associating Diagnostic Guidelines for Alzheimer’s Disease, the Petersen Criteria; or was defined by a standardised clinical measure, such as scores between 24-18 on the Mini-Mental State Exam (MMSE); scores ≤ 26 on the Montreal Cognitive Assessment (MoCA), scores between 15-27 on the St Louis University Mental Status (SLUMS), a Clinical Dementia Rating (CDR) score of 1 (for dementia) or 0.5 (for MCI); or a 4 (for dementia) or 3 (for MCI) on the Global Deterioration Scale (GDS). Studies which include a mix of participants with early dementia and MCI were included, however, studies which included healthy participants and participants with dementia at the later stages of the disease were excluded.”

We have reviewed our included studies to ensure that they still meet the inclusion criteria. We found one study which was no longer eligible.

Valdés, E. G., Andel, R., Lister, J. J., Gamaldo, A., & Edwards, J. D. (2019). Can Cognitive Speed of Processing Training Improve Everyday Functioning Among Older Adults With Psychometrically Defined Mild Cognitive Impairment?. *Journal of aging and health*, 31(4), 595-610.

We have removed this study from the analysis. However, this study only used 3 outcome measures (2 measure of cognition and 1 measure of activities of daily living), none of which were used by other studies included in this review. Therefore, it did not contribute much to the results.

As a result of removing this study:

- We have amended the study flow chart in Figure 1 to reflect that the new number of included studies is 91. This has also been amended in the abstract, results and discussion sections of the main manuscript.
- The new number of total outcome measures extract is 358. This has been amended in the abstract, results and discussion.
- We have removed this study from Table 1. This study was included in the tabulation of outcome measure domains by diagnosis in Table 2. This has now been removed.

Comment 4: It is not clear why the authors chose only to review studies of non-pharmacological interventions. There is no obvious reason why patients should want different outcomes from drug and non-drug interventions or why the best measures to assess these outcomes would vary with the nature of the intervention. This decision to limit the scope of the review should be justified.

While we agree that patients would want similar outcomes from both drug and non-drug treatment, it is important to consider how change occurs as a result of the intervention. Drug treatments tend to target biological mechanisms, while non-drug treatments do not. We acknowledge there is great deal of overlap in this area, both drug and non-drug treatments may aim to slow down or reverse the condition or modify symptoms. However, the mechanisms of how this change occurs should theoretically differ. Previous research has highlighted that the lack of focus on mechanism of change is a weakness in the evidence base of non-pharmacological treatments. We believe it is important the researchers in the area of research are clear on how change occurs, and which measures are best able to capture this.

We have included a description of this issue in the introduction (Page 5, lines 5-17):

“It is important researchers are clear on which domains their interventions are targeting, and which measures are best able to capture this change¹⁵. Pharmacological treatments target specific biological pathways underlying the disease; therefore, outcome measures have been chosen to reflect this and typically focus on cognitive and functional decline¹⁶. Non-pharmacological treatments generally do not target the underlying biological pathway of the disease therefore, outcome measures should theoretically differ between pharmacological and non-pharmacological treatments¹⁷. However, a review on non-pharmacological approaches to treating found that studies tended to pay little attention to the mechanisms of change underlying the intervention⁴. The expected mechanisms of change should affect which outcomes are used in non-pharmacological treatments for mild dementia and MCI.

In addition to being clear on how change arises in non-pharmacological treatments, there needs to be a more coherent use of outcomes and the measures used to capture these between studies to ensure a broad and robust evidence base¹⁵.”

We have also emphasised this point in the discussion (page 15, lines 10-23):

“There was great variability in the types of outcomes used to evaluate the different types of intervention. All studies measured cognition and all but one measured BPSD. A lack of clarity in how change occurs as a result of non-pharmacological treatments is a fundamental weakness in this area of work⁴. It is unlikely that all interventions being tested in this review could hope to improve cognition, however this is the most prevalent domain of outcome measures. There are a number of practical reasons as to why certain outcomes, and therefore outcome measures are used over others, In the past, pharmacological treatments have been required to include some measure of cognition,

functional or global assessment¹⁷, it is possible that this approach has influenced the choice in outcomes used in non-pharmacological studies. Furthermore, some measures may be used over others for more practical reasons. For example, measures which are short to administer and free to use may be priorities over others³¹. It is vital that outcome measures are selected depending on the domains the intervention is seeking to address³¹.

Comment 5: Similarly, it is not clear why studies of people with vascular cognitive impairments should be excluded. Many studies of non-pharmacological interventions and many MCI studies in particular do not try to make precise subtype diagnoses (and indeed the authors include studies which identify participants using simple clinical scales such as MMSE), so was it in fact possible to implement this exclusion criterion reliably? Why should key outcomes and the choice of outcome measure differ because the illness trajectory may be somewhat different? Indeed, this begs the question 'different from what'? Is the review intended to cover only mild dementia due to probable AD and probable AD-MCI? If so, this should be stated. Were studies of people with DLB, PDD and rarer dementias included or excluded? There are RCTs on, for example, cognitive training/rehab in people with MCI in PD which would appear to meet the inclusion criteria, but do not appear to have been included (e.g. Costa et al. Prospective memory performance of patients with Parkinson's disease depends on shifting aptitude: evidence from cognitive rehabilitation. *J Int Neuropsychol Soc* 2014; 20:717-26; Lawrence et al. Cognitive Training and Transcranial Direct Current Stimulation for Mild Cognitive Impairment in Parkinson's Disease: A Randomized Controlled Trial. *Parkinson's Disease* 2018; Article ID 4318475; Petrelli et al. Effects of cognitive training in Parkinson's disease: A randomized controlled trial. *Parkinsonism and Related Disorders* 2014; 20:1196-1202; and others).

Response 5: As the reviewer correctly pointed out, studies of Parkinson's disease dementia were excluded from this review. We have corrected this omission from the exclusion criteria. We agree that the use of outcome measures should not differ according to the trajectory of decline of the disease, and therefore have removed this justification from the exclusion criteria (Page 8, lines 2-3):

- The participants were diagnosed with vascular cognitive impairment or young-onset dementia or Parkinson's Disease Dementia

We agree that it is a limitation of this study that some sub-types of dementia and cognitive impairment (young-onset, Parkinson's disease dementia and vascular cognitive impairment) were excluded from this review. However, due to time constraints it was not possible to include these studies. We have included this as a limitation and recommendation for future research in the discussion (Page 16, lines 19-23):

"Due to time constraints, some sub-types of dementia and cognitive impairment (young-onset dementia, Parkinson's disease dementia and vascular cognitive impairment) were excluded from this review, which limits the applicability of these findings. Further research is needed to explore whether the pattern in the use of outcomes and outcome measures is similar in these groups, compared with the ones included in this review."

Comment 6: Lastly on the exclusion criteria, the authors should justify excluding studies intended to treat depression. Do they mean major depressive disorder or depressed mood / depressive symptoms? If the latter, then what is the reason for treating studies targeting this symptom domain differently from those targeting other non-cognitive symptoms?

Response 6: We were in fact excluding studies on participants with major depressive disorder. We have amended the inclusion criteria for greater clarity (P8 line 5):

- The intervention had the primary aim of treating major depressive disorder

Comment 7: A small point, but the title of Table 3 is incorrect. As the text correctly explains, this tabulates outcome domains (not outcome measures) against type of intervention. (The distinction

matters because the authors have argued in the Introduction that researchers should be careful to match the outcomes (i.e. outcome domains) they assess in their studies to the nature of the intervention).

We thank the reviewer for pointing this out, we have amended the title of this table. It now reads:

Table 3. Outcome measure domain by diagnosis and intervention

Comment 8: The authors' Conclusions also confuse outcomes and outcome measures in the third and fourth sentences.

We have carefully checked the entire manuscript for any points where outcomes and outcome measures have been confused and amended these.

Comment 9: The utility of the review would be greatly enhanced by additional information on the outcome measures. For example, which outcome measures have been validated in the population of interest? For which outcome measures is there any published information on a minimum clinically important difference? (MCID of outcome measures is a much-neglected aspect of outcome measures which is essential for interpreting research findings). Not including these features is to some extent a wasted opportunity to add to the literature in an important way and to influence the research agenda.

We agree with the reviewer that this is a very important area of future research, however this is outside of the scope of this review.

We had added this as a limitation to the present study (page 17, lines 2-5):

“As with the nature of scoping reviews, we are only able to present which outcome measures have been used in previous research, we are unable to draw conclusions as to which outcome measures should be used over others. Future research should explore which populations measures have been validated for and what constitutes a clinically useful change.”

Reviewer 2

Comment 1/ note 2: You may want to mention the inclusion of (or lack of) theoretical underpinning to a particular intervention. Line 12, page 5 “It is important researchers are clear on which domains their interventions are targeting, ...” alludes to this weakness. Theoretical underpinning for the intervention (if given at all) may indicate whether it is enacted as a treatment for an underlying driver of the disease, to slow progression and/or actually reverse the disease course, or whether it is enacted solely to modify symptoms. Very few interventions are theoretically grounded and evidence-based to address the underlying course of the disease. (on page 15, line 20 you do mention ‘theory’ so you have been thinking about it, just make more of it.)

We thank the reviewer for this helpful suggestion. We have amended the manuscript to emphasise the importance of the theory underlying non-pharmacological treatments. These sections have been described in the response to reviewer 1’s comment #4.

Comment 2: ‘care homes’ does not have an agreed international meaning - some consider nursing homes as care homes; Japan uses the term ‘group homes’... maybe long term care or an institutional setting or facility.... Not sure which to recommend, but just wanted to raise the point about universal nomenclature.

We thank the reviewer for bringing this to our attention. We have amended the inclusion criteria to (Page 7, line 23):

“Participants were living in long term care facilities or the community”

Comment 3: Some may question why you exclude papers not in English if you are aiming to look at outcome measures globally... obviously if you are funded to go deeper you would want to include these.

We agree that only including papers written in English is a limitation to our study. While we have removed the recommendation for global consistency in outcome measures (see reviewer 1 comment 1), we still think it is important to acknowledge this limitation. We have included this in our discussion (P16 lines 13-15).

Comment 4: Not sure about exclusion: ‘vascular cognitive impairment or young onset’ Did you have any that were excluded on this point? What does a different trajectory of decline have to do with measuring outcomes? (No action needed, just wondering)

We agree with the reviewer that trajectory of decline should not affect the use of outcome measures. We have addressed these issues in our response to reviewer 1’s comment #5.

Comment 6/note 6: You may want to add this recent reference for INTERDEM: Vernooij-Dassen, M., et al. (2019). "Bridging the divide between biomedical and psychosocial approaches in dementia research: the 2019 INTERDEM manifesto." *Aging Ment Health*: 1-7. 10.1080/13607863.2019.1693968.

Or this one from the JPND which has developed outcome measures: Oksnebjerg, L., et al. (2018). "Towards capturing meaningful outcomes for people with dementia in psychosocial intervention research: A pan-European consultation." *Health Expect* 21(6): 1056-1065. 10.1111/hex.12799

We thank the reviewer for bringing these studies to our attention.

Comment 7: Typos, grammar, clarity

We would like to thank the reviewer for their thorough reading of the manuscript and detailed suggestions. We have addressed each of the points raised.

Reviewer 2 Notes

Note 3: (Why so few studies in the UK...? No response required... just wondering.....)

Response 3: We agree it is strange that there are so few studies from the UK included in this review. Looking back at the studies which were excluded from this review, studies from the UK tended to focus on people with mild to moderate dementia. As studies including participants with moderate symptoms of dementia were excluded from this review, these studies were not eligible.

Note 4: Your comment is true but only a part of the picture. Sheehan (2012) gives good comparisons between various tools, raising points like the amount of time needed to administer, the expertise required of the person using it, and the availability of the tool in various languages. “it should be practical to use – in practice, this often depends on it being short (so it can be used in busy clinical practice or as an outcome measure in a trial such that participants are not overburdened by long interviews” The topic of outcome measurement tools is separate from which outcomes to measure, but is tied to it to some degree. The decision to measure a certain outcome is driven by perhaps a number of underlying issues. You may need to just add a paragraph in the discussion section about this so readers are not left with the impression that outcomes are being recommended with no understanding of the methodological implications. Did the studies shed any light at all on why certain tools were chosen over others....? I doubt it, but there may be a paper or two on this topic - not of which outcomes are recommended, but of which tools are used and why. Some tools are available on the web for free and others are under license. Economics and resource issues probably drive the use

of some tools over others. The MMSE might be the McDonald's of tools..... Cheap (or free), quick and available everywhere....!

We agree that there are pragmatic reasons as to why some tools are used over others. In practice, these pragmatic reasons may have a greater influence over which outcomes/outcome measures are used than the theoretical mechanisms of change. Due to the word count, we are unable to include a paragraph on this point. However, we have included a sentence on this in the discussion (Page 15, lines 18-20):

“Furthermore, some measures may be used over others for more practical reasons. For example, measures which are short to administer and free to use may be priorities over others³¹. It is vital that outcome measures are selected depending on the domains the intervention is seeking to address³¹.”

Note 5: You may want to mention the preponderance of multimodal interventions. These are important as cognitive impairment and dementia are recognized as multifactorial syndromes, so it takes multiple domains or modes to address it holistically. Various terms found in your reviewed papers were: multimodal, multidomain, multi-intervention, cognitive training and dynamic balance, memory enhancement training, physical and cognitive training, multicomponent exercise, multicomponent intervention and holistic health group. There were an average of 6.9 measures per study.

We agree that multi-component interventions should in theory use more outcome measures than other types of interventions. Therefore, we have added the following sentence to the discussion (P15, Lines 20-23):

“Several interventions in this review comprise of more than one component, e.g. physical activity and cognitive training. In these cases, it may take multiple measures over many domains to accurately capture change.”

Note 7: If your aim in the next phase of this work is that “more research is needed to highlight which outcome measures should be used over others” then perhaps add a sentence about how you might decide that. In other words, what do you see wrong with the ones in use.... Is it all about facilitating meta-analyses or is there more to it than that...?

We have included more details as what future research should consider when establishing which outcome and measures should be used over others (page 17, lines 2-5):

“As with the nature of scoping reviews, we are only able to present which outcome measures have been used in previous research, we are unable to draw conclusions as to which outcome measures should be used over others. Future research should explore which populations measures have been validated for and what constitutes a clinically useful change.”

Note 8: I dug through my Endnote and thought these studies may be of interest down the road....

Response 8: We greatly appreciate the time the reviewer has taken to suggest relevant studies and reviews. These have been helpful for the revision of the manuscript.

VERSION 2 – REVIEW

REVIEWER	Jenny McCleery Oxford Health NHS Foundation Trust, UK
REVIEW RETURNED	04-Feb-2020

GENERAL COMMENTS	The authors have paid close attention to the comments and have improved the manuscript, particularly in giving a more precise account of the scope of the work and a more detailed discussion of its limitations. The following minor points should be addressed:  1. The Abstract still refers to ‘early-stage dementia’ (under Eligibility sub-heading). 2. In the introductory paragraph on MCI, the authors should add that the conversion rate of approximately 10% is per annum. 3. The suggested papers related to Parkinson’s disease were about MCI. The added exclusion criteria should cover MCI in PD as well as PDD.
--

REVIEWER	Dr Garuth Chalfont Faculty of Health and Medicine Lancaster University Lancaster, UK
REVIEW RETURNED	31-Jan-2020

GENERAL COMMENTS	All changes have been reviewed and acceptance of the paper is now recommended (once minor typos below are fixed). #1 (Page 3, line 13) Do you mean ‘map trends in’... ? I think there is a word missing in this sentence. #2 (3, 36) Comma missing after ‘intervention’ #3 (5, 13) Period after ‘possible’ and capitalize ‘However’ #4 (5, 38) Period after ‘cognition’ and capitalize ‘Therefore’ #5 (16, 17) ‘profound effect of dementia’ #6 (16, 52) ‘outcome measures are selected’
---

VERSION 2 – AUTHOR RESPONSE

Reviewer comments	Action
Reviewer 1	
#1 (Page 3, line 13) Do you mean 'map trends in'... ? I think there is a word missing in this sentence.	We have amended this sentence to "map trends in"
#2 (3, 36) Comma missing after 'intervention'	We have made the suggested change
#3 (5, 13) Period after 'possible' and capitalize 'However'	We have made the suggested change
#4 (5, 38) Period after 'cognition' and capitalize 'Therefore'	We have made the suggested change
#5 (16, 17) 'profound effect of dementia'	We have amended this sentence to read "the profound effect of dementia"
#6 (16, 52) 'outcome measures are selected'	We have amended this sentence to read "outcome measures are selected"
Reviewer 2	
1. The Abstract still refers to 'early-stage dementia' (under Eligibility sub-heading).	We have amended this to read mild dementia
2. In the introductory paragraph on MCI, the authors should add that the conversion rate of approximately 10% is per annum.	This sentence now reads: Mild cognitive impairment (MCI) has been identified as a potential prodrome for dementia, with approximately 10% of people with MCI converting to a diagnosis of dementia per annum ⁸ .
3. The suggested papers related to Parkinson's disease were about MCI. The added exclusion criteria should cover MCI in PD as well as PDD.	This sentence now reads: "The participants were diagnosed with vascular cognitive impairment, young-onset dementia, Parkinson's Disease Dementia, or MCI with Parkinson's Disease"